# Intermediate and Transitory Inflammation Mediate Proper Alveolar Bone Healing Outcome in Contrast to Extreme Low/High Responses: Evidence from Mice Strains Selected for Distinct Inflammatory Phenotypes

**DOI:** 10.3390/biology13120972

**Published:** 2024-11-25

**Authors:** Priscila Maria Colavite, Michelle de Campos Soriani Azevedo, Carolina Fávaro Francisconi, Angélica Cristina Fonseca, André Petenucci Tabanez, Jéssica Lima Melchiades, Daniela Carignatto Passadori, Andrea Borrego, Marcelo De Franco, Ana Paula Favaro Trombone, Gustavo Pompermaier Garlet

**Affiliations:** 1Department of Biological Sciences, Bauru School of Dentistry, University of São Paulo, Al. Octávio Pinheiro Brisola, 9-75, Bauru CEP 17012-901, SP, Brazil; colavitepm@hotmail.com (P.M.C.); michelle_soriani@hotmail.com (M.d.C.S.A.); angelica_c_fonseca@hotmail.com (A.C.F.); danicarignatto@gmail.com (D.C.P.); 2Laboratory of Immunogenetics, Butantan Institute, Secretary of Health, Government of the State of São Paulo, Sao Paulo CEP 05503-900, SP, Brazil; andrea.borrego@butantan.gov.br (A.B.); mdf171717@gmail.com (M.D.F.); 3Pasteur Institute, Diagnostic Section, Sao Paulo CEP 01311-000, SP, Brazil; 4Department of Health Science, Universidade do Sagrado Coração, Bauru CEP 17011-160, SP, Brazil

**Keywords:** bone repair, inflammation, *Slc11a1*

## Abstract

The bone healing process involves the local development and resolution of an inflammatory process, which theoretically contributes to healing by mediating the migration of cells associated with the repair process, and to promote the local production of growth factors associated with osteogenic cellular differentiation. Using mice strains previously described to present distinct inflammatory behavior, namely AIRmin and AIRmax, which also present distinct alleles of the *Slc11a1* gene that also modulate inflammatory responsiveness, it is possible to investigate the association between inflammatory responsiveness and the bone healing outcome. After tooth extraction, it was observed that AIRmaxSS and AIRminRR presented the highest and lowest inflammatory readouts, respectively, associated with lowest repair levels in both strains. Conversely, the intermediate inflammatory phenotypes observed in AIRminSS and AIRmaxRR was associated with higher repair levels in such strains, supporting the idea that extreme high and low inflammatory responses are not ideal for a proper bone repair outcome, while an intermediate and transitory inflammation presented by AIRminSS and AIRmaxRR is associated with proper alveolar bone healing outcome.

## 1. Introduction

The bone tissue is a mineralized connective tissue whose functions include support, protection, and mineral reserve. Bone is periodically remodeled; this process is directly related to the dynamic balance of the bone formation activity by the osteoblasts and resorption by the osteoclasts [1,2].

Bone tissue presents a high capacity of regeneration when injured, for example, after fractures or teeth extraction, with the production of a new bone with morphofunctional characteristics similar to the original tissue at the injury site [1,3]. In this process, the local inflammatory immune reaction allegedly influences the outcome healing process [4,5]. Theoretically, a transient and moderate inflammatory process is supposed to mediate the migration of cells associated with the repair process and to promote the local production of growth factors associated with osteogenic cellular differentiation [5,6], while a chronic and exacerbated inflammatory response apparently impairs or delays the repair process [5,6]. However, while numerous studies have investigated the detrimental role of chronic inflammation over bone, little is known about the nature and extent of inflammatory host responsiveness required for proper bone healing; additionally, the exact molecular mechanisms involved in the immune and bone systems interplay in the healing process remains unknown.

At this point, another relevant factor must be considered in the overall analysis of the bone repair process. Macrophages and neutrophils play crucial roles in bone repair, with effective healing depending on the timely resolution of inflammation. Studies show that both the absence and excess of neutrophils disrupt the immune response, hindering repair. Bone healing begins with blood clot formation, inflammatory cell influx, and granulation tissue development, which is eventually replaced by bone [7]. Constructive inflammation is essential, as a controlled response speeds up healing, while inflammation blockade delays it. Chemotactic signals direct cells to lesion sites, where DAMPs activate inflammatory pathways that regulate neutrophil migration, supporting both inflammatory and regenerative functions [8].

In this context, our research group took advantage of a comparative analysis of the alveolar bone repair outcome in mice strains of genetically selected mice for maximal or minimal inflammatory reactions, namely AIRmax and AIRmin strains [9]. An increased inflammatory response associated with neutrophils and M1 macrophages is associated with the delay of bone repair in AIRmax mice, while a predominant M2 response is associated with a faster development of the healing process in the AIRmin strain [9]. Such data support the concept that variations in the intensity and nature of inflammatory immune response can modulate bone healing outcomes.

AIRmin and AIRmax strains have been characterized by the distinct inflammatory responsiveness and opposing outcomes of different pathological and healing processes [10]. The basis of the differential responsiveness of AIR strains involves variants of gene *Slc11a1* (“solute carrier family 11a member 1”), denominated as R or S alleles due its association with resistance (R) or susceptibility (S) of determined infectious diseases [11]. Subsequent studies demonstrated that R alleles predominate in AIRmax animals, while the presence of the allele S is characteristic of the AIRmin strain [12]. Further genotype-assisted breeding was performed to the generation with of homozygous mice the R and S alleles of the *Slc11a1* gene substrains, with the different AIRmin and AIRmax backgrounds, resulting in the AIRmaxRR, AIRmaxSS, AIRminRR and AIRminSS strains [13,14]. Such mice strains allow the analysis of R and S *Slc11a1* alleles and the AIRmin/AIRmax background on the inflammatory phenotype of substrains, as well of its subsequent influence in host response to infectious or healing processes [15].

It is important to say that the gene *Slc11a1* [16] has pleiotropic functions, acting primarily as a cation (such as Fe^2+^, Zn^2+^, Mn^2+^, and other divalent cations) transporter in macrophages [17], which plays prominent effects on macrophage activation and activity, including the production of nitric oxide TNF-α, IL-1 [18,19]. It is important to mention that such molecules, individually or in combination, are characteristics of M1 pro-inflammatory macrophages and can influence the nature and intensity of the immune inflammatory response and the subsequent outcome of alveolar bone repair. In this scenario, a timely and controlled neutrophil migration has also been considered an important element of the repair process [8,20]. Indeed, the combination of the *Slc11a1* S allele with the maximum inflammatory reaction background (i.e., AIRmaxSS strain) favors the ear tissue repair, while no tissue regeneration was observed in the animals with the same allele and the minimal inflammatory reaction (i.e., AIRminSS strain) [14], being the distinct healing outcome associated with different patterns of gene expression in each strain [21]. However, no studies were performed with such strains with the aim of studying the alveolar bone repair process [14,21,22].

In this context, we hypothesize that AIRmax and AIRmin mice with distinct *Slc11a1* R and S alleles can comprise an interesting model for the study of inflammatory host response interplay with the bone repair process. Therefore, in this study, AIRmin and AIRmax substrains of homozygous mice for *Slc11a1* R and S alleles were submitted for the extraction of the upper right incisor and comparatively evaluated regarding the intensity and nature of the inflammatory response and its effect on alveolar bone healing development and outcome by means of microtomographic, histological, birefringence and molecular analysis.

## 2. Materials and Methods

### 2.1. Animals

Experimental groups comprised 8-week-old AIRminRR, AIRminSS and AIRmaxRR, AIRmaxSS mice (strains generated and bred at Butantan Institute, Sao Paulo, Brazil), maintained during the experimental period in the animal facilities of the Department of Biological Sciences of FOB/USP. *Slc11a1* homozygous AIR sublines were produced by genotyping heterogeneous AIRmax and AIRmin mice with specific PCR primers for the *Slc11a1* SNP functional mutation at the 169 codons. Mice were mated in each line in order to obtain AIRmax and AIRmin mice presenting *Slc11a1* R and S alleles in homozygosis while maintaining their heterogeneous genetic background [13]. The sample size of 5 mice for each strain (for each time point) was determined based on previous studies performed with AIRmin/max substrains [9,14,21], to provide statistical power >90%. Throughout the period of the study, the mice were fed with sterile standard solid mice chow (Nuvital, Curitiba, PR, Brazil) and sterile water. The experimental protocol was approved by the local Institutional Committee for Animal Care and Use following the Guide for the Care and Use of Laboratory Animals principles (CEEPA-FOB/USP, processes # 003/2014).

### 2.2. Experimental Protocol and Mice Tooth Extraction Model

Animals were submitted to extraction of upper right incisor as previously described [23]. Male or female AIRminRR, AIRminSS and AIRmaxRR, AIRmaxSS mice (N = 5/time/group) were anesthetized by intramuscular administration of 80 mg/kg of ketamine chloride (Dopalen, Agribrans Brasil LTDA) and 160 mg/kg of xylazine chloride (Anasedan, Agribrands Brasil LTDA) in the proportion 1:1 determined according to the animal body mass. Importantly, animals presenting fractured teeth during the extraction were excluded from further analysis. At the end of the experimental periods (0-, 3-, 7- and 14-days after tooth extraction), the animals were killed with an excessive dose of anesthetic and the maxillae samples were collected. The maxillae samples were analyzed by micro-computed tomography (μCT), after the maxillae samples were dissected and prepared for histomorphometry, collagen birefringence and immunohistochemistry analysis or molecular analysis.

### 2.3. Micro-Computed Tomography (μCT) Assessment

The maxillae samples were scanned by the Skyscan 1174 System (Skyscan, Kontich, Belgium), at 50 kV, 800 μA, with a 0.5 mm aluminum filter and 15% beam hardening correction, ring artifacts, reduction, 180 degrees of rotation and exposure range of 1 degree. Images were captured with 1304 × 1024 pixels and a resolution of 14 μm pixel size. Projection images were reconstructed using the NRecon V1.6.9.8 software and three-dimensional images obtained by the CT-Vox 2.3 software. Morphological parameters of trabecular bone microarchitecture were assessed using the CTAn 1.1.4.1 software in accordance with the recommended guidelines [24]. A cylindrical region of interest (ROI) with an axis length of 3 mm (100 slices) and diameter of 1 mm was determined by segmenting the trabecular bone located from the coronal to apical thirds. Trabecular measurements analyzed included the tissue volume (TV), bone volume (BV), bone volume fraction (BV/TV, %), trabecular thickness (Tb.Th, mm), trabecular number (Tb.N, mm), and trabecular separation (Tb.Sp) [24,25]. The results were presented as the mean (±SD) for each evaluated structure. 

### 2.4. Histological Analysis

Serial sections (8 semi-serial sections of each maxilla, with a 5 μm thickness for each section) were obtained using a microtome (Leica RM2255, Wetzler, Germany) and stained with H.E. (hematoxylin and eosin). Morphometric measurements were performed by a single calibrated investigator with a binocular light microscope (Olympus Optical Co., Tokyo, Japan) using a 100× immersion objective and a Zeiss kpl 8X eyepiece containing a Zeiss II integration grid (Carl Zeiss Jena GmbH, Jena, Germany) with 10 parallel lines and 100 points in a quadrangular area. The grid image was successively superimposed on approximately 13 histological fields per histological section, comprised of all tooth sockets from the coronal limit adjacent to the gingival epithelium until the lower apical limit. For each animal/socket, sections from the coronal, medial, and apical thirds were evaluated. In the morphometric analysis, points were counted coinciding with the images of the following components of the alveolar socket: clot, inflammatory cells, blood vessels, fibroblasts, collagen fibers, bone matrix, osteoblasts, osteoclasts and other components (empty space left by the inflammatory exudate or intercellular liquid and bone marrow); similar to previous descriptions in other models [26,27,28]. The results were presented as the mean of volume density for each evaluated structure.

### 2.5. Picrosirius-Polarization Method

The picrosirius-polarization method and quantification of birefringent fibers were performed to assess the structural changes in the newly formed bone trabeculae matrix based on the birefringence of the collagen fiber bundles. Serial sections (8 semi-serial sections of each maxilla) with 5 μm thickness were cut and stained with picrosirius red stain; all sections were stained simultaneously to avoid variations due to possible differences in the staining process. Picrosirius red-stained sections were analyzed through a polarizing lens coupled to a binocular inverted microscope (Leica DM IRB/E), and all images were captured with the same parameters (the same light intensity and angle of the polarizing lens 90° to the light source). Adobe Photoshop CS6 software. Versão 13.0. Adobe, 2012 was used to delimit the region of interest (alveolar area comprised of new tissue with the external limit comprised of the alveolar wall). The quantification of the intensity of birefringence brightness was performed using the AxioVision 4.8 software (Carl Zeiss Microscopy GmbH, Jena, Germany). For quantification, the images were binarized for definition of the green, yellow and red color spectra and the quantity of each color pixels^2^ corresponding to the total area enclosed in the alveoli were measured. Mean values of 4 sections from each animal were calculated in pixels^2^.

### 2.6. Immunohistochemistry Analysis

Histological sections from 3, 7, and 14 days were deparaffinized following standard procedures. The material was pre-incubated with 3% hydrogen peroxidase block (Spring Bioscience Corporation, Pleasanton, CA, USA) and subsequently incubated with 7% NFDM to block serum proteins. The histological sections from all groups were incubated with anti-Gr1 polyclonal antibody (sc-168490) (Santa Cruz Biotechnology, Santa Cruz, CA, USA), anti-F4/80 polyclonal antibody (sc-26642) (Santa Cruz Biotechnology, Santa Cruz, CA, USA) anti-CD80 monoclonal antibody (sc-9091) (Santa Cruz Biotechnology, Santa Cruz, CA, USA) and anti-CD206 polyclonal antibody (sc-34577) (Santa Cruz Biotechnology, Santa Cruz, CA, USA) at 1:50 concentrations for 1 h at room temperature. The identification of the antigen–antibody reaction was performed using 3-3′-diaminobenzidine (DAB) and counter-staining with Mayer’s hematoxylin. Positive controls were performed in the mouse spleen for positive Gr1, F4/80, CD80, CD206, and receptors. The analysis of immunolabeled cells was performed by a single calibrated investigator with a binocular light microscope (Olympus Optical Co., Tokyo, Japan) using a 100× immersion objective and performed in a blinded way for groups and time points. The quantitative analysis for the different markers was performed throughout the alveolar extension. The absolute number of immunolabeled cells was obtained and subsequently used to calculate the immunolabeled cells mean for each mouse.

### 2.7. RealTime PCR Array Reactions

The extraction of total RNA from the remaining alveolus was performed with the RNeasyFFPE kit (Qiagen Inc., Valencia, CA, USA) according to the manufacturer’s instructions. The integrity of the RNA samples was verified by analyzing 1 mg of total RNA in a 2100 Bioanalyzer (Agilent Technologies, Santa Clara, CA, USA) according to the manufacturer’s instructions, and the complementary DNA were synthesized using 3 μg of RNA through a reverse transcription reaction (Superscript III, Invitrogen Corporation, Carlsbad, CA, USA). RealTimePCR array was performed in a Viia7 instrument (LifeTechnologies, Carlsbad, CA, USA) using a custom panel containing targets “Wound Healing” (PAMM-121), “Inflammatory cytokines and receptors” (PAMM-011) and “Osteogenesis” (PAMM-026) (SABiosciences, Frederick, MD, USA) for gene expression profiling. RealTimePCR array data were analyzed by the RT^2^ profiler PCR Array Data Analysis online software (SABiosciences, Frederick, MD, USA) for normalizing the initial geometric mean of three constitutive genes (GAPDH, ACTB, Hprt1) and subsequently normalized by the control group, and expressed as fold change relative to the control group.

### 2.8. Statistical Analysis

Data were presented as means ± SD; initially, the data distribution were tested by the Shapiro–Wilk normality test. Differences among data sets were statistically analyzed by ANOVA (followed by the Tukey test) or Kruskal–Wallis (followed by Dunn’s test), for the comparative analysis of the different time points in each strain, or student’s *t*-test or Mann–Whitney test for the comparative analysis of AIRmin and AIRmax substrains at individual time points. PCR array data were analyzed by the Mann–Whitney test followed by Benjamini–Hochberg test. Values of *p* < 0.05 were considered statistically significant. All were performed with GraphPad Prism 5.0 software (GraphPad Software Inc., San Diego, CA, USA).

## 3. Results

The results were presented performing a comparative analysis of alveolar bone repair features in substrains AIRminRR, AIRminSS, AIRmaxRR and AIRmaxSS. At this point, it is mandatory to consider that in view of the complexity of the combinatory analysis of four different groups, the analyses were performed from two viewpoints: one focused on the influence of the AIRmin/AIRmax inflammatory background (AIRminRR vs. AIRmaxRR and AIRminSS vs. AIRmaxSS comparisons), and the other focused on the influence of the R and S *Slc11a1* alleles (AIRminRR vs. AIRminSS and AIRmaxRR vs. AIRmaxSS comparisons) in the host response and bone healing readouts.

### 3.1. Micro-Computed Tomography μCT Analysis

Three-dimensional analysis from the μCT of the maxillae scanned in MicroCT (CT-Vox) showed the process comparatively alveolar bone healing in AIRminRR, AIRminSS, and AIRmaxRR, AIRmaxSS mice over time from 0, 3, 7, and 14 days after tooth extraction (Figure 1).

An overall qualitative analysis of the bone healing process demonstrated that in the initial period (0 h), the alveolus was completely void with no presence of hyperdense areas. At the 3 days timepoint, hyperdense areas became evident in all groups, with evidence of centripetal bone formation from the lateral and apical walls of the extraction sockets toward the center and the coronal region of the alveolus in both strains, followed by an increase in hyperdense regions along the subsequent experimental time points, which includes increasing trabecular number and thickness until the 14 d endpoint, where the alveoli appear to be filled with neo formed bone, characterizing a successful healing outcome (Figure 1A–D).

The quantitative analysis of μCT data confirms the qualitative three-dimensional analysis observations (Figure 2). In all the strains evaluated, the temporal analysis demonstrated a progressive increase along the time periods in the bone volume (BV), bone volume fraction (BV/TV), trabecular thickness (Tb.Th) and trabecular number (Tb.N) values, along a marked reduction in trabecular separation (Tb.Sp) from 14 days (Figure 2A–F). The comparative analysis focused initially on the influence of AIRmin/AIRmax on the inflammatory background, demonstrating that the AIRminRR strain presents decreased significant bone volume (BV) and bone volume fraction (BV/TV) (Figure 2B and Figure 2C, respectively), and increased significant trabecular separation (Tb.Sp) (Figure 2F) when compared to the AIRmaxRR; meanwhile, AIRminSS and AIRmaxSS showed no significant differences in analyzed parameters (Figure 2A–F). Regarding the influence of Scl11a1 alleles, AIRminRR presented a decreased significant number of trabeculae (Tb.N) (Figure 2E) and increased trabecular separation (Tb.Sp) (Figure 2F) when compared to AIRminSS; additionally, AIRmaxRR and AIRmaxSS showed no significant differences in analyzed parameters (Figure 2A–F).

Therefore, most divergent phenotypes were represented by higher bone formation levels in the AIRmaxRR strain, contrasting with the lower levels presented by the AIRminRR strain.

### 3.2. Histological and Histomorphometrical Evaluation

The histological and histomorphometrical analyses was performed to the qualitative and quantitative analysis, respectively, from characteristic element and distinct bone healing stages, such as, the presence of blood clot, inflammatory cells, fibroblasts, collagen fibers, blood vessels, bone matrix, osteoblasts, osteoclasts and other structures; the analyses reveal some differences between the strains (Figure 3 and Figure 4).

The initial overall histological analysis showed that in the period of 0 days after extraction of the right incisor tooth, the alveoli were filled by a blood clot in its full extension in all strains. At 3 days, all strains showed decreased blood clots and the presence of fiber, fibroblasts, and inflammatory cells in the alveolus margin region, demonstrating the onset of the repair process. At 7 days, the blood clot was gradually substituted by highly vascularized granulation tissue, characterized by many fibers and fibroblasts and the start of new bone formation from the margins of the socket; and at 14 days, the regions occupied by granulation tissue was gradually substituted by bone tissue in all strains (Figure 3A,B).

The quantitative histomorphometrical analyses, initially focused on the influence of AIRmin/AIRmax inflammatory background, demonstrated that AIRminRR presented initially (0 days) a lower density of other structures (exudate), followed a lower density of inflammatory cells, fibroblast, blood clot, other structures and connective tissue (3 d), while a higher density fibroblast, bone and inflammatory cells (7 d), followed by a decrease in the endpoint periods, and higher fibroblast and other structure density (14 d) levels when compared to AIRmaxRR. Additionally, we observed that AIRminSS presented a larger density of fibers, connective tissue, and inflammatory cells (3 d), followed by a lower density of blood clot, fibroblast, inflammatory cells, blood clot, other structures, and connective tissue (7 d), and decreased fibroblast and other structures in the endpoint (14 days) when compared to AIRmaxSS (Figure 4). 

On the other hand, when analyzing the influence of the RR/SS *Scl11a1* alleles, we observed that AIRminRR mice presented a large density of fibroblast, blood vessels, and inflammatory cells in the initial periods, followed (7 d) by a lower density of inflammatory cells and a larger density of fibroblast, and a larger density of fibers and inflammatory cells at the endpoint when compared to AIRminSS (Figure 4). Furthermore, AIRmaxRR presented a lower density of blood clots and a larger density of exudate (other structure) (0 d), followed by a larger density of other structure and inflammatory cells (3 d) and a larger density of blood vessels (7 d) and finally a lower density fibers and fibroblast, and a larger density of osteoblast at the endpoint when compared to AIRmaxSS (Figure 4).

Therefore, he divergent phenotype in the overall histomorphometric analysis is represented by the AIRmaxSS strain, which presented the lower densities of osteoblasts and bone matrix/tissue in parallel with higher inflammatory cell densities, while the other strains present more homogeneous histomorphometric readouts in general.

### 3.3. Collagen Birefringence Analysis

The analysis of the birefringence of the collagen fibers visualized through polarized and conventional light showed the presence of birefringence fibers in the color green, yellow, and red, with a predominance of red color in the final period; there were 14 in all strains (Figure 5A,B).

These data qualitatively represent different stages of fibers maturation during the experimental periods, with the onset of birefringence in the green spectrum, which indicates a less organized and more immature matrix, whereas fibers with a color spectrum varying from yellow to red were related to a matrix with a higher degree of organization and maturation. In relation, the kinetics of total area of collagen fibers analysis (pixels^2^) in the alveoli observed that all strains showed a significant increase (*p* < 0.05) in the period at 7 and 14 days (Figure 5C).

Comparing the influence of the AIRmin/AIRmax inflammatory background, we observed that the AIRminRR mice showed a significant decrease (*p* < 0.05) in the total number of collagen fibers in the periods of 7 and 14 days when compared to AIRmaxRR (Figure 5C), while AIRminSS and AIRmaxSS mice showed no significant differences. On the other hand, when analyzing the influence of *Scl11a1* alleles, we observed that AIRminRR and AIRminSS mice showed no significant differences, while AIRmaxRR mice showed a significant increase (*p* < 0.05) in the area of collagen fibers in the 7-day period when compared to AIRmaxSS (Figure 5C). Additionally, the analysis of the collagen fiber birefringence provided similar results to those observed in the total collagen fiber analysis (Figure 5D).

Therefore, most divergent collagen fiber phenotypes were represented by AIRmaxRR strains having higher levels, contrasting with the lowest levels presented by the AIRminRR strain.

### 3.4. Immunohistochemistry Analysis of Ly6g-GR1^+^, F4/80^+^, CD80^+^, CD206^+^

The immunohistochemical analysis was used to detect the presence of different inflammatory cells, such as granulocytes (Ly6g-Gr1^+^), macrophages (F4/80^+^), M1 macrophages (CD80^+^) and M2 macrophages (CD206^+^) in the sites of alveolar bone repair (Figure 6A–H). The quantitative analysis of immunostained cells comparing the AIRmin/AIRmax background inflammation showed that AIRminRR mice presented decreased counts of F4/80 (3 and 7 d) (Figure 7B) along with higher counts of CD80+ and CD206+ (3 and 14 d) (Figure 7C,D, respectively) when compared to AIRmaxRR. AIRminSS mice presented decreased counts of GR1+ (3, 7, 14 d) (Figure 7A), F4/80+ (3 d) and CD80+ (7 d) (Figure 7B,C, respectively) along and higher CD206+ counts (7 and 14 d) (Figure 7D) when compared to AIRmaxSS.

On the other hand, when analyzing the influence of *Scl11a1* alleles, we observed that AIRminRR mice presented decreased counts of GR1+ (3 d) and higher counts of CD206+ cells (3 and 7 d) (Figure 7A and Figure 7D, respectively) when compared to AIRminSS. AIRmaxRR presented decreased counts of GR1+ and CD80+ (3 and 7 d) (Figure 7A,C, respectively) along with higher F4/80+ (3 d) and CD206+ counts (7 and 14 d) (Figure 7B and Figure 7D, respectively) when compared to AIRmaxSS. Additionally, the results demonstrate that AIRmaxSS and AIRminRR present a similar M1/M2 ratio (approximately 1.5), while in AIRminSS and AIRmaxRR strains, there is a clear alteration in the M1/M2 balance (>3.0).

Therefore, most divergent phenotypes were represented by the higher granulocytes in parallel with lower M2 counts presented by AIRmaxSS strain contrasting with the opposite phenotype presented by the AIRminRR strain, while the AIRminSS and AIRmaxRR strains present intermediate and more balanced leukocyte counts in general.

### 3.5. Molecular Analysis Using Realtime PCRArray

Differential gene expression of several molecules involved in inflammatory response and bone healing (i.e., growth factors, immunological/inflammatory markers, extracellular matrix, and bone markers) was investigated employing a pool of samples from 3, 7 and 14 day periods analyzed using Real Time PCR array (Figure 8). The pooled analysis was chosen considering the complexity underlying the numerous targets, time points and experimental groups analysis. Similarly to the previous analysis, molecular data analysis was performed from two viewpoints, one focused on the influence of the AIRmin/AIRmax inflammatory background (when AIRminRR vs. AIRmaxRR and AIRminSS vs. AIRmaxSS comparisons are performed), and the other focused on the influence of *Slc11a1* alleles (via the AIRminRR vs. AIRminSS and AIRmaxRR vs. AIRmaxSS comparisons) in the bone healing process.

Initially, when analyzing the influence of the AIRminRR/AIRmaxRR inflammatory background, the results demonstrate that EGF, TGFB1, VEGFA and VEGFB expression were upregulated in the AIRminRR group in relation to the AIRmaxRR group; while FGF1 and FGF2 expression were downregulated in the AIRminRR group in compared in the AIRmaxRR (Figure 8A). Additionally, BMP2, BMP4, and BMP7 expression were downregulated in AIRminRR group compared to the AIRmaxRR group (Figure 8A). Considering the immunological markers, the expression of IL-1b, IL-6, TNF, and iNOS were downregulated in the AIRminRR compared to AIRmaxRR group, while CCR1, CCR2, CCR5, CXCR1, CCL2, CCL3, CCL5, CCL17, CCL9, CCL12, CCL17, CCL20, CCL25, CXCCL1, CXCL2, CXCL12, CXCL13 and CX3CL1 were downregulated in the AIRminRR group compared to AIRmaxRR (Figure 8A,B). Furthermore, extracellular matrix and bone markers, including MMP1A, MMP2, MMP9, TIMP1, CD106, CD166, OCT-4, NANOG, CD44, CD34, CD73, CD146, NES, CD133, Runx2, Alpl, Phex, Sost, RANKL and RANK were downregulated in AIRminRR compared to AIRmaxRR (Figure 8C).

When the molecular data were analyzed to determine the influence of the AIRmin/AIRmax inflammatory background, the results demonstrate that FGF1 and FGF2 expression were downregulated in the AIRminSS group in relation to AIRmaxSS, while BMP2 expression was upregulated in AIRminSS (Figure 8A). Considering the immunological markers, the expression of IL-1b, IL-6 and TNF were downregulated in the AIRminSS compared to the AIRmaxSS group, while IL-10 and ARG2 were upregulated in the AIRminSS group (Figure 8A). Additionally, the results demonstrate that CCR1, CCR2, CXCR1, CCL2, CCL9, CCL20, CXCL1, CXCL2 and CXCL12 were downregulated in the AIRminSS group compared AIRmaxSS, while CX3CL1 was upregulated in the AIRminSS group (Figure 8B). The analysis of extracellular matrix and bone markers reveals that MMP1A, MMP2, MMP8, MMP9, CD166, CD73 and CTSK were downregulated in AIRminSS compared to the AIRmaxSS groups, while CD133 and Runx2 were upregulated in AIRminSS (Figure 8B).

Complementarily, when the influence of *Slc11a1* alleles was analyzed, the results demonstrate that VEGFA and VEGFB expression were upregulated in the AIRminRR group in relation to the AIRminSS group, in parallel with downregulated BMP2, BMP4, and BMP7 expression in the AIRminRR group (Figure 8A). Considering the immunological markers, the expression of IL-1b, IL-6, TNF, and iNOS were downregulated in the AIRminRR compared to the AIRminSS group, in parallel with CCR1, CCR2, CCR5, CXCR1, CCL2, CCL5, CCL12, CCL17, CCL20, CCL25, CXCCL1, CXCL2, CXCL12, CXCL13 and CX3CL1 downregulation in the AIRminRR group (Figure 8A,B). Regarding extracellular matrix and bone markers, MMP1A, MMP2, MMP9, CD106, CD166, OCT-4, NANOG, CD44, CD34, CD73, CD146, NES, CD133, CD105, Runx2, Alpl, Phex, Sost, RANKL and RANK were downregulated in AIRminRR compared to the AIRminSS group (Figure 8C).

When AIRmaxRR vs. AIRmaxSS where compared, VEGFB expression was upregulated in the AIRmaxRR group in relation to the AIRmaxSS group, along upregulated levels of BMP2 and BMP4 in the AIRmaxRR group (Figure 8A). Considering the immunological markers, the expression of IL-1b, IL-6, IL-10, IL-17, TNF, CCR1, CCR2, CCR5, CXCR1, CCL2, CCL20 and CXCL2 were downregulated in the AIRmaxRR group compared to AIRmaxSS group, while ARG2 CXCL12, CXCL13 and CX3CL1 were upregulated in the AIRmaxRR group (Figure 8A,B). Regarding extracellular matrix and bone markers, MMP1A, MMP2, MMP8, MMP9 and MMP13 were downregulated in AIRmaxRR compared to the AIRmaxSS groups, while TIMP1, TIMP3, CD106, CD44, CD146, Runx2, Phex, Sost and RANKL were upregulated in AIRmaxRR (Figure 8C).

The molecular analysis reveals the AIRminRR and AIRmaxSS strains to have the most divergent phenotypes, given the expression of pro-inflammatory cytokines and chemokines were downregulated in the AIRminRR strain while they were upregulated in AIRmaxSS. Intermediate inflammatory phenotypes were observed in the AIRminSS and AIRmaxRR strains, associated with upregulated MSCs and bone formation markers.

## 4. Discussion

The exact role of the inflammatory immune response in the bone healing process is still unclear, but a favorable outcome of the alveolar bone healing process is theoretically associated with a moderate/balanced and transitory inflammatory response, while insufficient or exacerbated responses seem to have a detrimental influence on the healing process. In this context, this study took advantage of mice strains with distinct inflammatory responsiveness derived from opposing inflammatory backgrounds (i.e., AIRmin and AIRmax) and selected *Slc11a1* alleles (R and S alleles) to further investigate the interplay between host inflammatory responsiveness and bone healing. Initially, it is important to consider that the analysis confirms the distinct inflammatory phenotypes in AIRminRR, AIRminSS, AIRmaxRR, and AIRmaxSS, revealing distinct healing outcomes in such strains. From the inflammatory viewpoint, AIRmaxSS presented the highest inflammatory readouts (i.e., inflammatory cell counts and inflammatory mediators expression), while AIRminRR represents the opposing lowest inflammatory phenotype. It is noteworthy that both strains presented the lowest repair levels (bone density and volume), supporting the idea that extremely high and low inflammatory responses are not ideal for a proper bone repair outcome. Accordingly, the intermediate inflammatory phenotypes observed in AIRminSS and AIRmaxRR were associated with higher repair levels in such strains.

In this context, the analysis of AIR substrains homozygous for *Slc11a1* alleles allows the analysis of the influence of R and S alleles in a given biological process (such as bone repair in this specific case), and also allow the analysis of the influence of other genetic factors composing the ‘min’ and ‘max’ genetic backgrounds. Therefore, it is possible to analyze and discuss the data obtained from two viewpoints, one focused on the influence of the AIRmin/AIRmax inflammatory background (when AIRminRR vs. AIRmaxRR and AIRminSS vs. AIRmaxSS comparisons are performed), and another focused on the influence of both R and S *Slc11a1* alleles (via the AIRminRR vs. AIRminSS and AIRmaxRR vs. AIRmaxSS comparisons) in the bone healing process. It is noteworthy that while previous studies investigated the alveolar bone repair in AIRmax and AIRmin strains, which present different proportions of R and S alleles, with the predominance of the R allele in the AIRmax strain and the S allele in the AIRmin strain [12], the mixed genetic composition does not allow the exact analysis of R/S alleles independently. However, such comparisons do not clearly show better/worse inflammatory backgrounds and *Slc11a1* alleles associated with extreme healing phenotypes, in contrast with a previous study focused on critical ear defects [14]. The initial studies focused on ear hole defects demonstrate that AIRmax presented a full regeneration while AIRmin did not present ear hole regeneration capacities [14,21]. Subsequent studies demonstrated that mice presenting the AIRmin background mice did not regenerate ear holes irrespective of carrying RR/SS *Slc11a1* alleles, while AIRmaxRR also presented a defective regeneration and only AIRmaxSS could properly regenerate the ear holes [14].

However, instead of dichotomic regeneration/non-regeneration phenotypes [14], our data reveal a more complex interaction between ‘min’ and ‘max’ backgrounds and R and S *Slc11a1* alleles in the determination of inflammatory phenotypes, and the subsequent impact of such phenotypes in the bone healing outcome. Accordingly, genetic studies identified numerous QTLs associated with the distinct inflammatory responsiveness derived from AIRmin and AIRmax backgrounds, independently of the *Slc11a1* alleles [4]. Among them, six inflammatory QTL located on chromosomes 1, 7, 8, 12, 14, and 16 are implicated in the differential tissue repair/regeneration phenotype in AIRmax and AIRmin mice [21]. Furthermore, it is mandatory to consider that bone and ear (cartilage) defects are different and involve distinct stages and cell types, which could account for the distinct healing phenotypes observed [9,14,23]. Indeed, while ear defects can be considered critical, whose regeneration only occurs in very specific conditions, alveolar bone defects seem to be less critical and to repair in a less restrictive way, despite significant differences in the timing and amount of bone formation between the different strains investigated [9]. In this context, our results demonstrate that distinct combinations of ‘min’ and ‘max’ genetic backgrounds with *Slc11a1* alleles result in the better bone repair readouts, which are coincident with the development of more balanced inflammatory responses presented by AIRminSS and AIRmaxRR strains. While it is not possible to define absolute extreme dichotomic healing phenotypes, the analysis of host response feature in the different mice strains reveals interesting insights into the putative mechanisms mediating inflammatory responsiveness interplay with bone healing outcomes.

Initially discussing the extreme inflammatory phenotypes, AIRmaxSS presented the highest inflammatory readouts (i.e., inflammatory cell counts and inflammatory mediators’ expression), and consequently presented the lowest repair levels (bone density and volume). In this context, macrophages are thought to contribute to both repair [29,30] and remodeling [31,32], initially promoting pro-inflammatory effects associated with M1 polarization, followed by a phenotypic shift towards M2 dominance, which is considered anti-inflammatory and pro-repair [33,34,35,36]. The initial predominance of M1 macrophages in healing sites has been considered as primordial for the activation of acute inflammatory response, in view of its pro-inflammatory features, being able to produce high levels of pro-inflammatory cytokines like TNF, IL-6 and INOS [37]. Accordingly, the increased macrophage migration is supported by the expression of the chemokines CCL2 and CCL5, and the chemokine receptors CCR2 and CCR5, characteristically involved in macrophages chemotaxis [38]. However, AIRmaxSS strain response involves a persistence of M1 response in detriment of the usual M2 dominance in late repair stages, which could account for the limited repair observed in such strain. Accordingly, while AIRmaxSS and AIRminRR present a similar and relatively balanced M1/M2 ratio, there is a clear alteration in the M1/M2 balance in AIRminSS and AIRmaxRR strains. Indeed, M2 macrophages are considered favorable for regenerative outcomes [39,40,41,42], characterized by high expression of anti-inflammatory and pro-reparative cytokines as IL-10, TGFb1 and ARG-1 [34,43,44].

In addition to the theoretical role exerted by macrophages in the bone repair process, it is important to consider that patterns of neutrophil response diverge within the AIRmin/max substrains evaluated. Initially we must consider that the AIRmaxSS strain response involves increased migration and prolonged presence of neutrophils, associated with increased levels of neutrophil-related chemokines, such as CXCL1 and CXCL2, likely driven by a dominant pro-inflammatory response that may contribute to limited repair [9,14]. In fact, the resolution of the inflammatory process, including neutrophil clearance via apoptosis or retrotaxis, is an essential step in healing [45]. Importantly, neutrophil chemotactic recruitment, when timely and coordinated, is associated with pro-inflammatory activity as well as anti-inflammatory, immunoregulatory, and reparative properties [8,20]. Accordingly, the limited repair observed in the AIRminRR strain is associated with decreased neutrophil migration, decreased levels of CXCL1 and CXCL2, reinforcing that such leukocyte subset can present constructive roles in the bone repair process. Indeed, recent studies further suggest that both the exclusion of neutrophils and neutrophilia can compromise repair, disrupting the natural inflammatory immune response development [46,47]. Moreover, the nonspecific blockade of inflammation slows repair, supporting the concept of constructive inflammation [3,31].

Conversely, AIRminRR represents the opposing lowest inflammatory phenotype, comprising a limited expression of pro-inflammatory and M1 mediators. This initial M1 response is supposed to contribute to the chemoattraction of different cell types required for the proper repair [30,34,35,36]. Interestingly, the AIRminRR presents a generalized downregulation of chemokines and chemokines expression, which could account for the lower leukocyte infiltration. Furthermore, this reduced expression of chemotactic cues includes CXCL12, a major chemotactic factor for MSCs [48], whose marker expressions are also downregulated at healing sites in this strain. It is also mandatory to consider the overall decrease in angiogenic and growth factor expression, including key osteogenic inducers such as BMPs, observed in the AIRminRR strain. Such elements are usually upregulated in healing sites under control conditions [48] and considered as essential factors for proper cell proliferation and differentiation along the repair process [9,48]. While the repair process involves an initial M1 polarization that rapidly evolves to M2 in control conditions [9], and M2 cells allegedly contribute to tissue repair, it is possible to consider that the limited M1 response critically limits the cell migration, which in turn can account for the ineffective repair observed in this strain despite the upregulation of M2 marker expression in AIRminRR.

Interestingly, previous studies demonstrate a similar endpoint healing in AIRmax and AIRmin strains and that the delayed repair observed in the AIRmax strain was associated with increased presence of neutrophils and M1 macrophages [9]. However, despite the delayed repair, a late M1 to M2 switch seems to compensate or overcome the initial pro-inflammatory dominance, resulting in the similar endpoint healing mentioned [9]. Conversely, the phenotypes observed in AIRmaxSS and AIRminRR do not seem to include similar compensatory mechanisms, comprising a sustained and exacerbated response (AIRmaxSS) and an insufficient initial pro-inflammatory response (AIRminRR), which differ from the response described to the original AIRmax and AIRmin [9]. At this point, it is possible to consider that non-resolving pro-inflammatory phenotypes could benefit from immunomodulatory strategies, such as using FTY720, VIP, and PACAP, demonstrated to favor the development of the M2 phenotype in the alveolar bone repair process with an increase in the bone tissue in the treated groups [49,50].

Apart from the opposing extreme inflammatory phenotypes, the intermediate inflammatory phenotypes observed in AIRminSS and AIRmaxRR were associated with higher repair levels in such strains, supporting the idea that a balanced host response is required for a proper healing outcome. Indeed, while opposing extreme levels of chemokines and chemokine receptors were observed in AIRminRR and AIRmaxSS, such elements are expressed in intermediate levels in AIRminSS and AIRmaxRR strains, which could account for the overall intermediate presence of inflammatory cells in the healing sites of such strains. Additionally, healing-related chemokines, such as CCL20, CCL25, and CXCL4 [9], are notably upregulated in such strains, in parallel with high levels of growth and osteogenic factors and MSCs markers. Accordingly, bone repair involves a localized inflammatory immune response, which is believed to aid the healing process through the recruitment of repair-associated cells, the production of growth factors, and osteogenic cellular differentiation [23,51,52,53,54]. While blocking the inflammatory process delays bone healing, chronic and excessive inflammatory responses are also linked to impaired healing [23,51,52,53,54,55,56,57,58]. This suggests that carefully regulating the nature, extent, and intensity of the host response at the injury site is crucial for a successful bone healing outcome.

At this point, another relevant factor must be considered in the overall analysis of bone repair processes. In addition to genetic and cellular factors, environmental influences like chronic alcohol exposure must also be considered, as they significantly impact bone repair. Studies show that chronic alcohol consumption compromises bone repair in humans and animal models by disrupting critical pathways necessary for bone homeostasis [59]. Persistent and episodic alcohol intake alters cellular signaling pathways essential for bone regeneration, potentially amplifying or diminishing the inflammatory responses crucial to effective healing [59,60]. Further research indicates that prolonged ethanol exposure increases TNF-alpha expression in osteoblast-like cells, thereby creating an environment less conducive to cellular proliferation and regeneration [61]. These findings collectively highlight that both genetic predispositions and environmental factors, such as chronic alcohol use, can intensify inflammatory responses and impair bone repair. This underscores the necessity for a balanced inflammatory response for optimal bone regeneration and the potential benefits of therapeutic strategies aimed at modulating excessive inflammation to improve healing outcomes.

It is also mandatory to consider that most of the studies focused on the interplay between host inflammatory immune response and bone repair comprise experimental studies in rodents. While there are obvious differences in the bone repair process in humans and rodents, rats and mice have been proven to allow the recapitulation of the bone repair stages described in humans and, therefore, comprise valuable models in the context of the ethical and technical limitations to perform similar studies in patients. The alveolar bone healing process is a complex and organized sequence of events involving different cell types and tissues, as well as significant interactions with the host immune system [62,63]. Unlike endochondral healing, which has been widely studied in fracture models, alveolar bone healing, an intramembranous process, is still poorly understood, particularly regarding the immune response. Following tooth extraction, the healing process initiates with blood clot formation, providing the foundation for repair, which is progressively replaced by granulation tissue, marked by fibroblast and endothelial cell proliferation, as well as new blood vessel formation, which is essential for bone tissue regeneration [28,64,65,66]. This provisional tissue, rich in inflammatory cells, forms an initial extracellular matrix that gradually matures into thick bone trabeculae with well-defined medullary canals, signaling the onset of bone remodeling as osteoblasts and osteoclasts coordinate to stabilize the newly formed bone [67,68]. Given these insights, experimental models using mice are invaluable for investigating this repair process due to their well-documented inflammatory responses and the availability of various strains, including genetically modified ones, allowing extensive experimental exploration. 

Our results support the overall concept that a moderate inflammatory process is necessary for the success of alveolar bone repair, while insufficient or exacerbated/non-resolving inflammation is detrimental to the repair outcome. Considering that the inflammatory immune response is present along all the stages with distinct features and that the mice strains used in the study are characterized by distinct inflammatory and immunological responsiveness, it is possible to consider that inflammatory immune factors act as modulators of other cell types actions along the repair process. In this context, considering that a cytokine can modulate the expression of a chemokine that attracts MSCs, and MSCs can be further modulated by the local environment in parallel with distinct M1/M2 polarization and increased/decreased migration of neutrophils, it is difficult to clearly establish the exact nature of each factor role in the repair outcome, and further studies are required to determine the individual role of the modulated factor in the whole process. Our data also reveal a complex interaction between ‘min’ and ‘max’ backgrounds and R and S *Slc11a1* alleles in the determination of inflammatory phenotypes, which do not clearly evidence better/worse inflammatory backgrounds and *Slc11a1* alleles associated with extreme healing phenotypes but reveal features of inflammatory gradients associated with better/worse repair outcomes.

## 5. Conclusions

In summary, the present study demonstrated showed that extremely low and high inflammatory phenotypes, presented, respectively, by AIRminRR and AIRmaxSS strains, are associated with limited alveolar bone healing, while intermediate and transitory inflammation presented by AIRminSS and AIRmaxRR are associated with proper alveolar bone healing outcomes.

## Figures and Tables

**Figure 1 biology-13-00972-f001:**
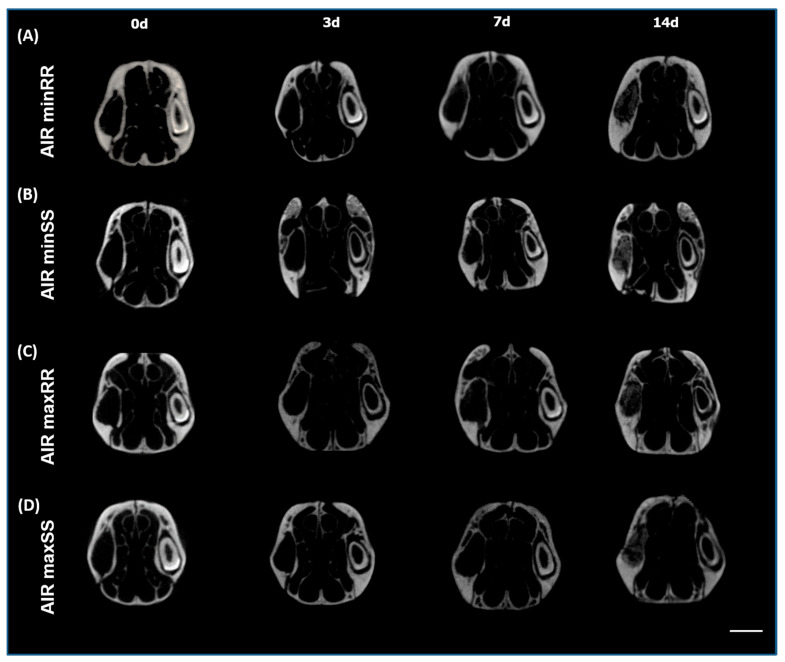
Micro-computed tomography (μCT) analysis of bone healing process kinetics in AIRminRR, AIRminSS and AIRmaxRR, AIRmaxSS (**A**–**D**). Samples from 8-week-old male or female mice were scanned with the μCT system (Skyscan 1174; Skyscan, Kontich, Belgium): evaluated at 0, 3, 7, and 14 days post-tooth extraction to evaluate the kinetics of the bone healing process. Images were reconstructed using the NRecon V1.6.9.8 software and three-dimensional images were obtained with the CT-Vox 2.3 software. The sectioned maxilla is represented at the coronal planes. The delimited area represents the bone healing process kinetics in mice.

**Figure 2 biology-13-00972-f002:**
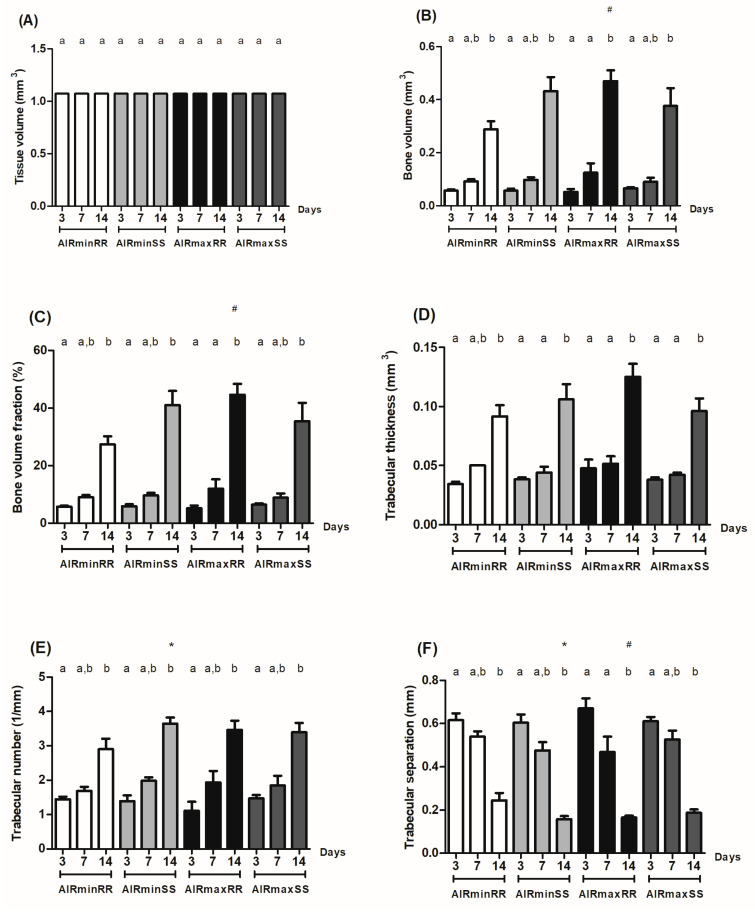
Morphological parameters of the trabecular bone microarchitecture in AIRminRR, AIRminSS, AIRmaxRR, and AIRmaxSS mice. Data were assessed using the CTAn 1.1.4.1 software from the cylindrical region of interest (ROI) determined by segmenting the trabecular bone located from the coronal to apical thirds. (**A**) Trabecular measurements analyzed included the tissue volume (TV), (**B**) bone volume (BV), (**C**) bone volume fraction (BV/TV %), (**D**) trabecular thickness (Tb.Th, mm), (**E**) trabecular number (Tb.N, mm), and (**F**) trabecular separation (Tb.Sp). * indicates significant statistical differences (*p* < 0.05) between the AIRminRR vs. AIRminSS and AIRmaxRR vs. AIRmaxSS groups, # indicates significant statistical differences (*p* < 0.05) between the AIRminRR vs. AIRmaxRR and AIRminSS vs. AIRmaxSS groups and different letters indicate significant statistical differences (*p* < 0.05) between the periods.

**Figure 3 biology-13-00972-f003:**
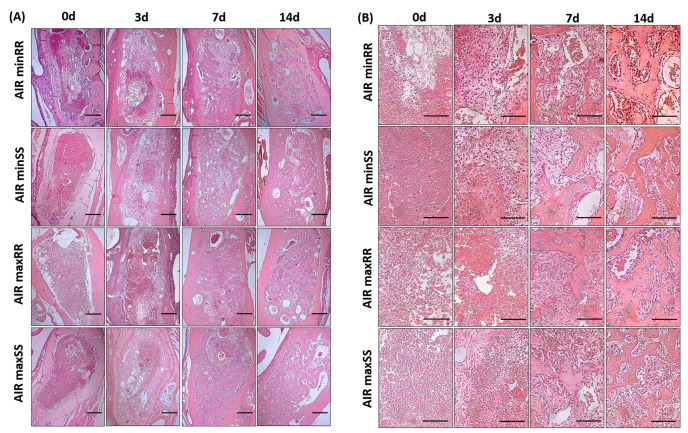
Histological aspects of the medial and apical thirds from tooth sockets in the bone healing process. Representative sections of the alveolar bone healing kinetics at 0-, 3-, 7- and 14-days post-extraction of the upper right incisor in AIRminRR, AIRminSS, AIRmaxRR, and AIRmaxSS mice. HE staining, (**A**) original magnification 10× and (**B**) original magnification 40×. Bar = 100 μm.

**Figure 4 biology-13-00972-f004:**
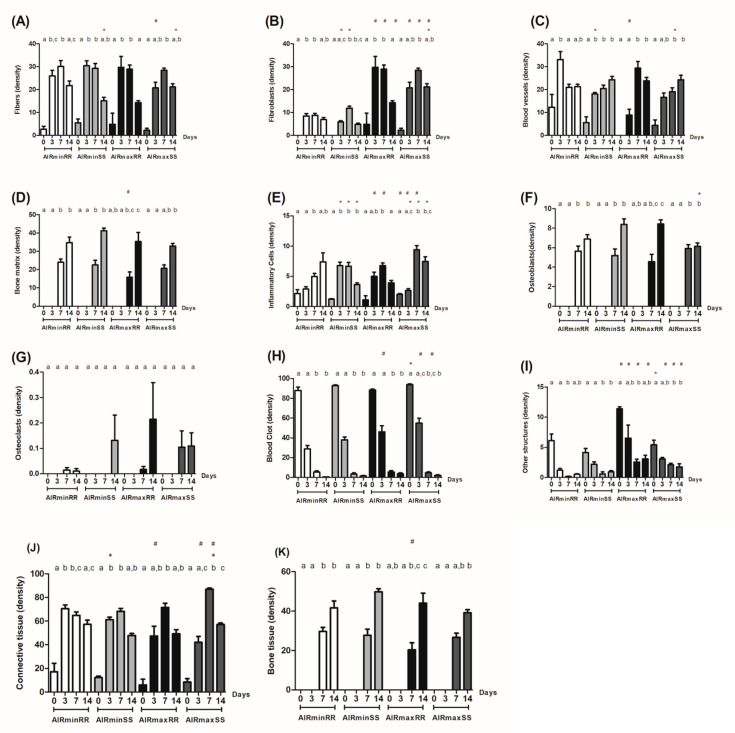
Histomorphometric analysis of alveolar bone healing kinetics after tooth extraction. Results are presented as the means (±SD) of density for each structure of the alveolar socket. (**A**) fiber, (**B**) fibroblasts, (**C**) blood vessels, (**D**) bone matrix, (**E**) inflammatory cells, (**F**) osteoblast, (**G**) osteoclasts, (**H**) blood clot, (**I**) other components (empty space left by the inflammatory exudate or intercellular liquid), (**J**) total density of connective tissue (represented by the sum of fibers, fibroblasts, blood vessels, and inflammatory cells) and (**K**) bone tissue (represented by the sum of its structural components bone matrix, osteoblasts and osteoclasts). * indicates significant statistical differences (*p* < 0.05) between the AIRminRR vs. AIRminSS and AIRmaxRR vs. AIRmaxSS groups, # indicate significant statistical differences (*p* < 0.05) between the AIRminRR vs. AIRmaxRR and AIRminSS vs. AIRmaxSS groups and different letters indicate significant statistical differences (*p* < 0.05) between the periods.

**Figure 5 biology-13-00972-f005:**
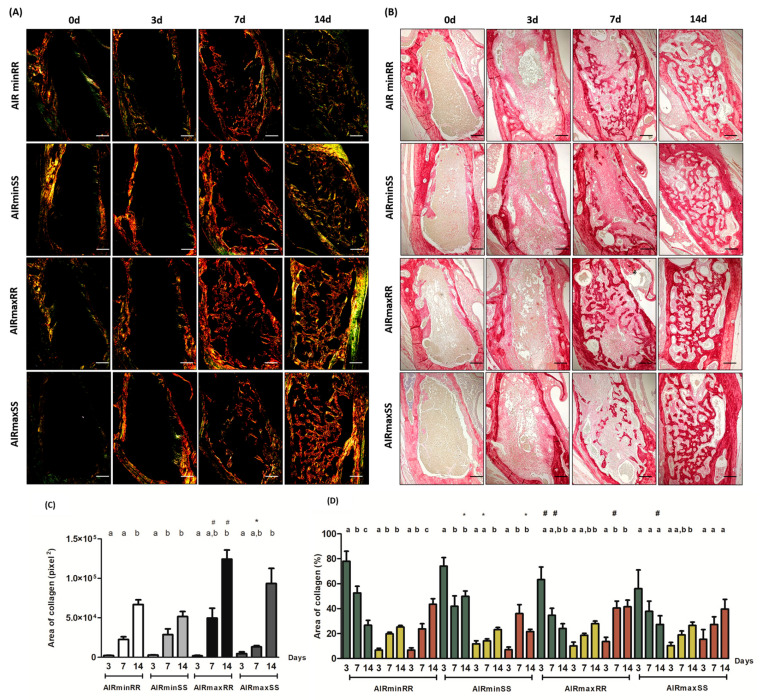
Quantification of birefringent fibers by the picrosirius-polarization method in the bone healing process after tooth extraction. Representative sections of picrosirius red staining visualized upon polarized (**A**,**B**) to identify collagen fibers types at 0, 3, 7 and 14 days post-extraction of the upper right incisor. Green birefringence color indicates thin fibers; yellow and red colors in the birefringence analysis indicate thick collagen fibers. Original magnification 10×. Bar = 100 μm. Intensity of birefringence was measured with the Image-analysis software (AxioVision 4.8 softwareCarl Zeiss, Oberkochen, Germany) for identification and quantification; (**C**) total area of collagen fibers (pixel^2^) and (**D**) area of collagen from each birefringence color (%). Results are presented as the mean (±SEM) of pixels^2^ for each color in the birefringence analysis in the bone healing at 0-, 3-, 7- and 14-days post-extraction. * indicates significant statistical differences (*p* < 0.05) between the AIRminRR vs. AIRminSS and AIRmaxRR vs. AIRmaxSS groups, # indicates significant statistical differences (*p* < 0.05) between the AIRminRR vs. AIRmaxRR and AIRminSS vs. AIRmaxSS groups and different letters indicate significant statistical differences (*p* < 0.05) between the periods.

**Figure 6 biology-13-00972-f006:**
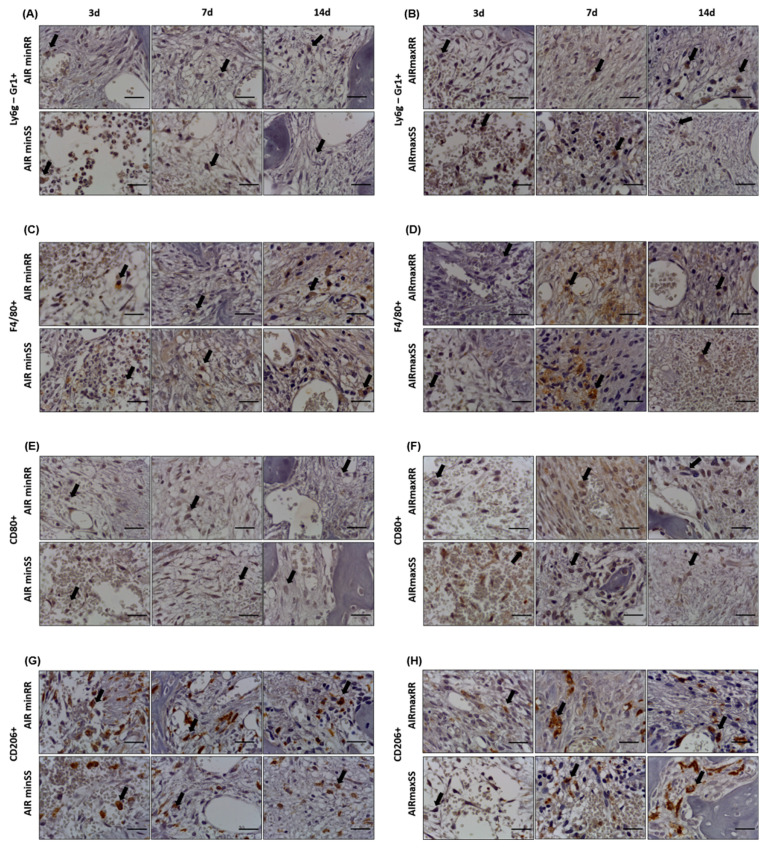
Immunohistochemistry analysis for (**A**,**B**) GR1+, (**C**,**D**) F4/80+, (**E**,**F**) CD80+ and (**G**,**H**) CD206+ cells present in the bone repair process in the AIRminRR, AIRminSS and AIRmaxRR, AIRmaxSS mice. Representative sections from medial thirds of the socket at days 3, 7 and 14 days after tooth extraction. Indirect staining MACH4+ DAB, anti-staining Mayer hematoxylin; objective of 100×. The arrows indicate the immunolabeled cells.

**Figure 7 biology-13-00972-f007:**
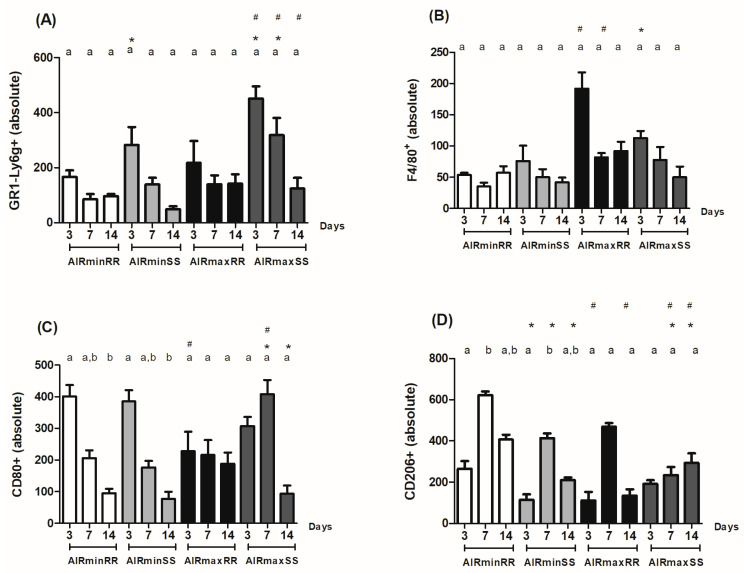
Analysis of inflammatory cells in the alveolar bone healing kinetics after tooth extraction in AIRminRR, AIRminSS and AIRmaxRR, AIRmaxSS. (**A**) Immunohistochemistry quantification corresponding to Ly6g-Gr1+ immunolabelled with Ly6g-Gr1+, (**B**) F4/80+ immunolabelled with anti-F4/80+, (**C**) CD80+ immunolabelled with anti-CD80+ and (**D**) CD206+ immunolabelled with anti-CD206. Results are presented as the means (±SEM) and * indicates significant statistical differences (*p* < 0.05) between the AIRminRR vs. AIRminSS and AIRmaxRR vs. AIRmaxSS groups, # indicates significant statistical differences (*p* < 0.05) between the AIRminRR vs. AIRmaxRR and AIRminSS vs. AIRmaxSS groups and different letters indicate significant statistical differences (*p* < 0.05) between the periods.

**Figure 8 biology-13-00972-f008:**
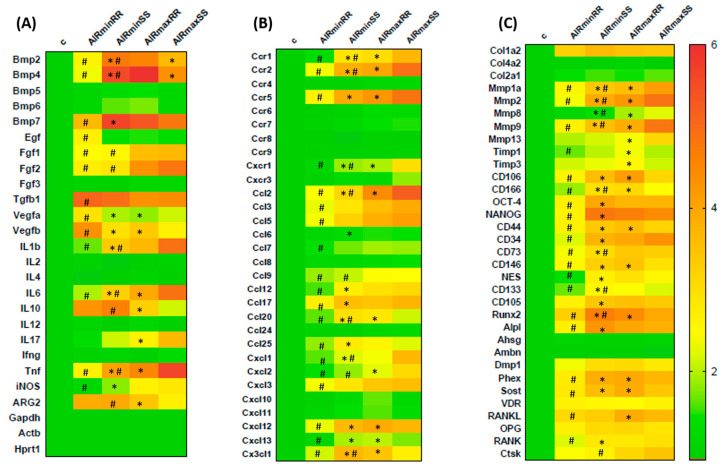
Molecular analysis (PCRArray) using a heat map to quantify the expression of (**A**) growth factors and cytokine markers, (**B**) extracellular matrix markers and (**C**) bone markers in bone healing process among AIRminRR, AIRminSS and AIRmaxRR, AIRmaxSS, post-extraction. PCRArray of pooled samples. Results were obtained when comparing the relative expression of the different groups to the normalizing control and * indicates significant statistical differences (*p* < 0.05) between the AIRminRR vs. AIRminSS and AIRmaxRR vs. AIRmaxSS groups, # indicates significant statistical differences (*p* < 0.05) between the AIRminRR vs. AIRmaxRR and AIRminSS vs. AIRmaxSS groups.

## Data Availability

The data that support the findings of this study are available from the corresponding author, [GPG], upon request.

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
