# Peer review of "Intermediate and Transitory Inflammation Mediate Proper Alveolar Bone Healing Outcome in Contrast to Extreme Low/High Responses: Evidence from Mice Strains Selected for Distinct Inflammatory Phenotypes"

_biology, 2024, doi:10.3390/biology13120972_

Round 1
Reviewer 1 Report
Comments and Suggestions for Authors
Colavite et al. have examined the effects of inflammatory genotypes (AIRmin/max) and the S/R alleles of the transporter Slc11a1 on alveolar bone regeneration in a genetically manipulated mouse model. A great deal of complementary data are presented on tooth regeneration at different times after removal including structural (microCT/histological), immunohistochemical (specific staining of inflammatory cell types) and mRNA expression (PCR arrays) consistent with the hypothesis that genotypes with maximal or minimal inflammatory phenotypes have inferior bone regeneration relative to intermediate phenotypes. They conclude that some inflammatory pathways are required for optimal bone regeneration.
The data sets appear comprehensive and appropriately analyzed from the statistical perspective. However, several concerns remain relative to study limitations, originality and lack of perspective with regards to work in the overall area of inflammation and bone regeneration in other bone types.
1. The authors should discuss how the current work differs from and provides new information relative to their previous publication in the area (reference 8). In particular the significance or lack thereof of the S/R Slc11a1 genotype in relation to phenotype.
2. Given the large number of gene expression differences observed between the different strains, how can the authors distinguish those differences driving the phenotypes observed and those that contribute little?
3. Limitations to the current study should be discussed. In particular differences between mouse and human tooth formation and the small N.
4. There is an excess of self-referencing and a lack of reference to other groups studies on the influence of inflammation on bone regeneration. Excess self referrals should be removed eg. Refs 6, 10 or 11, some of 27-32, 58 or 59 or they should be justified. References to other studies in this area for example of TNF/IL1 signaling in relation to fracture healing and distraction osteogenesis by Lumpkin CK Jr et al. and studies on fracture healing by Jon Callaci and colleagues to provide a wider perspective on the role of inflammation on bone formation.
Author Response
Comments and Suggestions for Authors
Colavite et al. have examined the effects of inflammatory genotypes (AIRmin/max) and the S/R alleles of the transporter Slc11a1 on alveolar bone regeneration in a genetically manipulated mouse model. A great deal of complementary data are presented on tooth regeneration at different times after removal including structural (microCT/histological), immunohistochemical (specific staining of inflammatory cell types) and mRNA expression (PCR arrays) consistent with the hypothesis that genotypes with maximal or minimal inflammatory phenotypes have inferior bone regeneration relative to intermediate phenotypes. They conclude that some inflammatory pathways are required for optimal bone regeneration.
The data sets appear comprehensive and appropriately analyzed from the statistical perspective. However, several concerns remain relative to study limitations, originality and lack of perspective with regards to work in the overall area of inflammation and bone regeneration in other bone types.
Answer: The authors are extremely thankful to the reviewers for the energy and time invested in reviewing this manuscript. We appreciate the detailed and constructive feedback that allowed us to improve the quality of manuscript. All modifications performed in order to fulfill the reviewers demands are highlighted in the revised version of the manuscript (highlighted in yellow).
- The authors should discuss how the current work differs from and provides new information relative to their previous publication in the area (reference 8). In particular the significance or lack thereof of the S/R Slc11a1 genotype in relation to phenotype.
Answer: The current work, titled "Intermediate and Transitory Inflammation Mediate Proper Alveolar Bone Healing Outcomes in Contrast to Extreme Low/High Responses: Evidence from Mice Strains Selected for Distinct Inflammatory Phenotypes," analyzes the R and S alleles of Slc11a1 and the AIRmin/AIRmax background in the inflammatory phenotype of the substrains (AIRminRR, AIRminSS, AIRmaxRR, and AIRmaxSS), as well as their subsequent influence on the host response to infectious or healing processes. We clarify that the previously cited work, Colavite et al., 2018 examined only the mouse strains designated AIRmax and AIRmin. While AIRmax and AIRmin strains present different proportions of R and S allele, with the predominance of R allele in the AIRmax strain and the S allele in the AIRmin strain, the mixed genetic composition does not allow the exact analysis of R/S alleles independently. Therefore, by means of genotype-assisted breeding, homozygous mouse strains for the R and S alleles of the Slc11a1 gene were generated, with the different genetic backgrounds of AIRmin and AIRmax, resulting in the AIRmaxRR, AIRmaxSS, AIRminRR, and AIRminSS strains (Biguetti et al, 2018; Garlet at al., 2010; Takayanagi 2005). The differences with the previous studies were added to discussion section as requested.
- Given the large number of gene expression differences observed between the different strains, how can the authors distinguish those differences driving the phenotypes observed and those that contribute little?
Answer: That’s a very important and complex question. Considering the existence of multiple stages of bone healing process, and the involvement of numerous mediators in each stage, in a large scale analysis such as the one performed by RealTimePCRarray in this study evidences a large number of genes whose expression is modulated. Considering that the inflammatory immune response is present along all the stages with distinct features, and that the mice strains used in the study are characterized by distinct inflammatory and immunological responsiveness, the discussion have a main focus in inflammatory immune factors, considered to act as modulators of other cells types actions along the repair process. In this context, considering that a cytokine can modulate the expression of a chemokine that attracts MSCs, and MSCs can be further modulated by the local environment, in parallel with distinct M1/M2 polarization and increased/decreased migration of neutrophils (just to cite an example) it is difficult to clearly stablish the exact nature of each factor role in the repair outcome. In the view of the reviwer question, we added sentences to the discussion section considering such complexity inherent to the repair process.
- Limitations to the current study should be discussed. In particular differences between mouse and human tooth formation and the small N.
Answer: We clarify that the experimental N, comprising 5 mice for each strain (5for each time point) was determined based in previous studies performed with AIRmin/max substrains (Colavite 2018), to provide statistical power >90%. It is also important to consider that the genetic selection used to generate AIRminRR, AIRminSS and AIRmaxRR, AIRmaxSS mice strains results in a relatively small variation whitin each strain, similar to that observed in isogenic strains, allowing the use of a relatively low sample size to generate a proper statistical power, which is also in line with the ethical recomdations/guidelines for animal experimentation.
Regarding the reviewer request for considering ‘differences between mouse and human tooth formation’, we assume that the reviewer requests a discussion between mouse and human bone repair. Thefore, we added to the discussion section sentences focused in the similarities and differences between mouse and human bone repair and the value of experimental models of bone repair, in order to fulfill the reviewer request.
- There is an excess of self-referencing and a lack of reference to other groups studies on the influence of inflammation on bone regeneration. Excess self referrals should be removed eg. Refs 6, 10 or 11, some of 27-32, 58 or 59 or they should be justified. References to other studies in this area for example of TNF/IL1 signaling in relation to fracture healing and distraction osteogenesis by Lumpkin CK Jr et al. and studies on fracture healing by Jon Callaci and colleagues to provide a wider perspective on the role of inflammation on bone formation.
Answer: As requested, we removed some the self-references; we clarify that some were maintained since provide essential links and background to the present study without the possibility of being replaced by studies with the same content, and some are referent to previous studies that provide methodological information and support. Also in the view of the reviewer request, we included the requested citations, which provided a broader perspective on the role of inflammation in bone formation.
REFERENCE
Biguetti C.C., Vieira A.E., Cavalla F., Fonseca A.C., Colavite P.M., Silva, R.M., Trombone A.P.F., Garlet G.P. CCR2 Contributes to F4/80+ cells migration along intramembranous bone healing in maxilla, but its deficiency does not critically affect the healing outcome, Front Immunol. 9 (2018) 1804.
Garlet G.P. Destructive and protective roles of cytokines in periodontitis: a re-appraisal
from host defense and tissue destruction viewpoints, J. Dent. Res. 89 (12) (2010) 1349–1363.
Takayanagi H. Inflammatory bone destruction and osteoimmunology, J. Periodontal Res. 40 (4) (2005) 287–293.
Colavite, P. M., A. E. Vieira, C. E. Palanch Repeke, R. P. de Araujo Linhari, R. De Andrade, A. Borrego, M. De Franco, A. P. F. Trombone and G. P. Garlet. Alveolar bone healing in mice genetically selected in the maximum (AIRmax) or minimum (AIRmin) inflammatory reaction. Cytokine.2019,114,47-60.

Reviewer 2 Report
Comments and Suggestions for Authors
The authors produced a thorough research paper titled 'Intermediate and Transitory Inflammation Mediate Proper Alveolar Bone Healing Outcomes in Contrast to Extreme Low/High Responses: Evidence from Mice Strains Selected for Distinct Inflammatory Phenotypes.' The paper addresses a clear research question, investigating how varying levels of host immune response affect bone regeneration, particularly in the alveolar bone of the jaw. The authors employed a well-established mouse model with known immune response levels and used a robust set of methodologies to evaluate bone healing and profile immune responses. The conclusions drawn are consistent with the results, indicating that a balanced or intermediate immune response yields better healing outcomes than either an insufficient or an exaggerated immune response.
The significance of these findings suggests potential therapeutic avenues for modulating host response—at least locally—to enhance bone healing in clinical settings. It may also provide a foundation for predicting surgical outcomes through inflammatory marker screening.
A minor comment regarding Figure 1: the representative μCT images do not appear to be selected at a consistent plane for group comparison. If possible, further explanation or adjustment would improve clarity.
Comments on the Quality of English LanguageOverall, the language is understandable. However, certain sentences need revision for clarity and improved flow, particularly in the following lines:-
- Line 104
- Line 236
- Line 295
Author Response
Comments and Suggestions for Authors
The authors produced a thorough research paper titled 'Intermediate and Transitory Inflammation Mediate Proper Alveolar Bone Healing Outcomes in Contrast to Extreme Low/High Responses: Evidence from Mice Strains Selected for Distinct Inflammatory Phenotypes.' The paper addresses a clear research question, investigating how varying levels of host immune response affect bone regeneration, particularly in the alveolar bone of the jaw. The authors employed a well-established mouse model with known immune response levels and used a robust set of methodologies to evaluate bone healing and profile immune responses. The conclusions drawn are consistent with the results, indicating that a balanced or intermediate immune response yields better healing outcomes than either an insufficient or an exaggerated immune response.
The significance of these findings suggests potential therapeutic avenues for modulating host response—at least locally—to enhance bone healing in clinical settings. It may also provide a foundation for predicting surgical outcomes through inflammatory marker screening.
Answer: The authors are extremely thankful to the reviewers for the energy and time invested in reviewing this manuscript. We appreciate the detailed and constructive feedback that allowed us to improve the quality of manuscript. All modifications performed in order to fulfill the reviewers demands are highlighted in the revised version of the manuscript (highlighted in yellow).
A minor comment regarding Figure 1: the representative μCT images do not appear to be selected at a consistent plane for group comparison. If possible, further explanation or adjustment would improve clarity.
Answer: All μCT images were selected in the coronal plane, as referenced in the figure legend, following the standards mentioned in the Materials and Methods section.
Comments on the Quality of English Language
Overall, the language is understandable. However, certain sentences need revision for clarity and improved flow, particularly in the following lines (Line 104, 236, 295):
Answer: The sentences were corrected as requested.

Reviewer 3 Report
Comments and Suggestions for Authors
I primarily focused on the immunological aspects of the study.
Major Comments:
1. For enhanced clarity in all immunostaining figures, I recommend indicating the immunopositive cells with arrows or other markers, as this would facilitate reader interpretation of the staining results.
2. Inflammation involves various immune cell types beyond M1 macrophages. The manuscript lacks a clear rationale for focusing solely on macrophages; it would benefit from an explanation addressing why other immune cells are not considered or how the chosen focus aligns with the study’s overall objectives.
3. The current analysis of inflammatory status could be strengthened by evaluating the ratio of pro-inflammatory (M1) to anti-inflammatory (M2) cells rather than focusing solely on the absolute numbers of individual cell types. Co-staining and calculating the relative proportions of these cell populations would provide a more comprehensive indication of the inflammatory balance.
4. Aside from Figure 8, the study predominantly relies on immunostaining data. Including complementary cellular data would substantively enhance the robustness of the findings, providing a stronger foundation for the conclusions drawn.
Minor Comments:
- Please ensure alignment of all figures for improved visual consistency across the manuscript.
Author Response
Reviewer 03
Comments and Suggestions for Authors
I primarily focused on the immunological aspects of the study.
Answer: The authors are extremely thankful to the reviewers for the energy and time invested in reviewing this manuscript. We appreciate the detailed and constructive feedback that allowed us to improve the quality of manuscript. All modifications performed in order to fulfill the reviewers demands are highlighted in the revised version of the manuscript (highlighted in yellow).
Major Comments:
- For enhanced clarity in all immunostaining figures, I recommend indicating the immunopositive cells with arrows or other markers, as this would facilitate reader interpretation of the staining results.
Answer: The figures were revised as requested. We included the arrows to indicate the immunolabeled cells.
- Inflammation involves various immune cell types beyond M1 macrophages. The manuscript lacks a clear rationale for focusing solely on macrophages; it would benefit from an explanation addressing why other immune cells are not considered or how the chosen focus aligns with the study’s overall objectives.
Answer: We agree with the reviewer viewpoint and in fact the discussion was quite biased towards macrophages roles. It is important to consider that macrophages are highly prevalent in repair sites in quantitative terms, and specially after the discovery of M1 and M2 subsets, macrophages have been in the spotlight of repair-focused studies. Therefore, most of the recent literature in the field focus in the macrophages theoretical roles in the repair process. However, despite a minor focus in the recent literature, neutrophils are also potentially involved in repair process, and in the original version of the discussion section, the proper attention to the neutrophil-related data were not given. In the view of the important reviewer concern the discussion section was revised.
- The current analysis of inflammatory status could be strengthened by evaluating the ratio of pro-inflammatory (M1) to anti-inflammatory (M2) cells rather than focusing solely on the absolute numbers of individual cell types. Co-staining and calculating the relative proportions of these cell populations would provide a more comprehensive indication of the inflammatory balance.
Answer: We agree with the reviewer viewpoint and the calculation of M1/M2 ratio was performed. The results demonstrate that AIRmaxSS and AIRminRR present a similar M1/M2 ratio (approximately 1.5), while in AIRminSS and AIRmaxRR strains there is a clear alteration in the M1/M2 ratio (>3.0), reinforcing the concept that a balanced inflammatory immune response is required for a proper repair. Such information was included in the revised version of the manuscript and were considered in the discussion section.
Regarding the possible co-staining we clarify that despite the efforts to co-stain macrophages with multiple markers the results were not considered satisfactory from the technical aspect and consequently were not added to the manuscript.
- Aside from Figure 8, the study predominantly relies on immunostaining data. Including complementary cellular data would substantively enhance the robustness of the findings, providing a stronger foundation for the conclusions drawn.
Answer: We agree with the reviewer viewpoint that additional data would provide a more robust support to the finding. However, as previously mentioned, co-staining attempts were not considered satisfactory from the technical aspect and consequently were not added to the manuscript. Also, our group tried to perform flow cytometry-based cellular characterization, as previously performed with periodontal tissues samples. However, due to the requirement of additional tissue processing due the features of bone (and forming bone) tissue (the samples require stronger enzymatic and mechanical processing when compared with periodontal tissues to allow the ‘cell release’ from matrix), it was not possible to recover cells in number and quality (specially viability) enough to the flow cytometer analysis.
Minor Comments:
- Please ensure alignment of all figures for improved visual consistency across the manuscript.
Answer: The figures were corrected as requested.

Round 2
Reviewer 1 Report
Comments and Suggestions for Authors
The authors have revised their submission adequately and the result is much improved.
Reviewer 3 Report
Comments and Suggestions for Authors
The authors have addressed most of my comments, I have no further comments.